# Young Australian Women’s Views on Peer Support for Self-Harm: A Qualitative Study

**DOI:** 10.3390/ijerph22121874

**Published:** 2025-12-17

**Authors:** Amy Wang, Demee Rheinberger, Samantha Tang, Helen Christensen, Alison L. Calear, Katherine Boydell, Alexis Whitton, Aimy Slade, Anastasia Hronis

**Affiliations:** 1Discipline of Clinical Psychology, Graduate School of Health, University of Technology, Ultimo, NSW 2007, Australia; amyzlwang@hotmail.com; 2Black Dog Institute, University of New South Wales, Randwick, NSW 2031, Australia; d.rheinberger@blackdog.org.au (D.R.); samantha.tang@blackdog.org.au (S.T.); h.christensen@blackdog.org.au (H.C.); k.boydell@blackdog.org.au (K.B.); a.whitton@blackdog.org.au (A.W.); aimy.slade@blackdog.org.au (A.S.); 3Faculty of Medicine and Health, University of New South Wales, Kensington, NSW 2033, Australia; 4Centre for Mental Health Research, The Australian National University, Canberra, ACT 2601, Australia; alison.calear@anu.edu.au

**Keywords:** peer support, self-harm, self-injurious behaviour, intentional self-harm, non-suicidal self-injury, self-harm recovery

## Abstract

Self-harm among young women has been rising internationally and in Australia, yet many are not in contact with formal services. Peer support may play an important role in managing self-harm; however, it remains under-investigated. This qualitative study explored how young Australian women perceive peer support for managing self-harm. Using purposive sampling, twenty-seven women (*M* = 20.9, *SD* = 2.1) with lived or living experiences of self-harm participated in semi-structured interviews. Data were analysed using reflexive thematic analysis. Five themes highlighted perceived benefits and risks of peer support: 1. Affirmation—peer support offers understanding and validation that reduce shame and stigma; 2. Connection to community—shared experience reduces isolation and supports learning; 3. Empowerment—peer support promotes hope, autonomy, and help-seeking; 4. Capacity matters—limited psychological knowledge and emotional resources can constrain or burden peers; 5. Perception can be distorted—in un-moderated online contexts, normalisation, glamorisation, and competitive dynamics of self-harm may increase risk. These findings offer insight into how young women understand the different aspects of peer support for self-harm and may inform the development of structured and moderated peer support options. Future research should focus on the design and evaluation of safe and effective peer support initiatives.

## 1. Introduction

Peer-based support services are emerging as a promising avenue for young people who self-harm [1]. Despite the growing prevalence, the role of peer support in helping individuals manage self-harm remains under investigated, with limited evidence regarding its benefits, risks, and mechanisms of action.

Self-harm, defined as intentionally causing harm to oneself irrespective of suicidal intent, has become increasingly prevalent among young people globally [2]. Methods of self-harm vary and include cutting, burning, hitting, and starving oneself, and ingesting harmful substances [3]. The rising rates of self-harm, particularly among young women, are alarming. In Australia, the number of hospitalisations for self-harm in women aged 15–19 has nearly doubled, from 374 to 689 per 100,000 individuals between 2008 and 2021 [4], while non-hospitalised instances are estimated to be ten times higher [5]. While the current study centres on young women, evidence indicates that young men also face significant and distinct challenges related to self-harm. Research has shown that young men often encounter different help-seeking barriers, rely on alternative coping mechanisms, and may experience self-ham within gender contexts that shape their willingness to disclose distress [6]. A recent scoping review further illustrated the complexity of men’s self-harm, noting how their distress is expressed and supported [7]. These studies underscore the need for gender-sensitive research on self-harm; however, exploring men’s self-harm experiences lies beyond the scope of the current study.

Self-harm presents significant public health concerns, including associations with suicide risk and various mental health difficulties [8,9]. Despite these risks, many young people who engage in self-harm do not seek professional help [2,10]. Societal and self-imposed stigma often prevent young people from accessing clinical services [11,12], with recent evidence indicating that Australians delay accessing mental health support by an average of 12 years [13]. In addition, systemic gaps in mental health services, such as long waiting lists, limited availability of professional support, and sub-optimal engagement outcomes with clinicians, leave many without timely care [14,15]. Furthermore, although recommended interventions such as Dialectical Behavioural Therapy and Cognitive Behavioural Therapy have demonstrated effectiveness in reducing self-harm behaviours [16], reviews indicate that the treatment effects remain small [17,18]. Studies have also found that young people prefer support from peers, as an alternative or supplementary source of emotional and practical assistance [1,19].

Peer support has become a vital part of mental health care services and is increasingly integrated into formal and informal therapeutic frameworks [20,21]. Peer support refers to a system where individuals with shared experiences of mental illness or distress provide emotional, social, and practical support to each other [20,22]. In the broader literature, peer support can take several forms. Informal peer support refers to naturally occurring, unstructured exchange of emotional and/or practical assistance within friendship, peers, or community networks. In contrast, formal peer support involves trained peer workers who operate within defined roles, supported by supervision and protocols [23]. In addition, peer support may also occur in digital environments that vary substantially in structure. More specifically, moderated online spaces include supervision aimed at reducing harmful content, whereas unmoderated online spaces allow unrestricted user-generated interactions for support opportunities [24]. Although these distinctions provide conceptual clarify, for the purposes of the current study, peer support is broadly defined as any support offered outside clinical environments, either face-to-face or through online communities, by individuals who have lived or living experience of self-harm.

Peer support has been shown to offer emotional and practical benefits for individuals who engage in self-harm. One of the most commonly reported advantages is a reduction in feelings of isolation [1,24]. Sharing experiences in a non-judgmental environment fosters a sense of belonging and validation, helping individuals feel understood and less alone [23,25,26,27]. This shared understanding can also promote self-awareness, as discussing experiences enables individuals to articulate their thoughts and feelings, and identify triggers, which facilitates personal insights and growth [23,28,29]. In addition, peer settings provide opportunities to exchange practical coping strategies, such as distraction and sensory techniques, which support emotional regulation and empower individuals to take more control of their self-harm behaviours [30,31]. Importantly, peer support also promotes hope and optimism by providing success stories of people who are further along in their recovery [25,32].

Despite the many benefits, peer support is not without its risks. A key concern is that un-moderated or poorly structured peer environments, particularly online, can normalise or even reinforce self-harm behaviours [24,33]. Comparison dynamics may also arise, when individuals evaluate their experiences against others and feel pressured to engage in more severe self-harm to validate their struggles [31,34,35]. Another potential risk is the exposure to triggering content, as discussions or images shared in online space can provoke distress or self-harm urges [1,26,31,36,37]. However, some research suggests that exposure to severe depictions of self-harm may deter rather than encourage engagement, possibility by highlighting its harmful consequences [38,39]. The relationship between content exposure and self-harm behaviour appears intricate and is likely influenced by individual factors, such as the individual’s emotional resilience, or their stage of recovery.

While there is growing literature on the impact of peer support for self-harm, limited research has explored this topic in the Australian context. Brasier et al. [40] found that peer support workers provide benefits such as offering emotional comfort, reducing feelings of isolation and distress, and de-escalating crisis, to individuals attending emergency departments in Australia due to mental distress, which may include self-harm presentations. Another Australian study [41] identified that youth peer support roles promote self-acceptance and interpersonal skill development, but also involve challenges such as emotional strain, boundary management, role ambiguity, and lack of recognition from clinical staff. However, both studies primarily focused on the perspectives of formal, paid peer support workers, rather than informal, unpaid peer support provided by young peers within their personal or community networks. To date, no studies have examined the role of peer support specifically in the context of self-harm in Australia, from the perspective of young people with lived experiences.

Given the growing uptake of peer support in mental health and increasing self-harm rates among young Australian women, it is important to understand how this demographic perceives peer support interactions. This study aims to address a critical gap in the literature by investigating the views of young Australian women on peer support for self-harm, exploring their experiences and concerns. This study seeks to provide insights that will inform the development of safe and effective peer support strategies that can complement existing mental health services.

## 2. Materials and Methods

### 2.1. Study Setting

This qualitative study forms part of a larger research initiative investigating young Australian women’s experiences with self-harm which will be published across numerous studies (e.g., [42]). Ethical approval was granted by the University of New South Wales (UNSW) Human Research Ethics Committee (HC230437).

### 2.2. Recruitment

Participants were recruited between October and November 2023 using purposive sampling via advertisements on the Black Dog Institute’s website and social media accounts, including Facebook and Instagram. Sample size (*N* = 27) was determined based on the concept of “information power” which suggests that a small- to medium-sized sample is sufficient to answer broad questions about an uncommon phenomenon while maintaining depth necessary to understand the variation between different experiences [43]. Given the focused research aim, purposive sampling, and rich data generated during interviews, a sample of 27 participants was considered adequate to support meaningful and nuanced analysis.

Potential participants visited the study website, where they could access the Participant Information Statement and Consent Form outlining the study procedures and reimbursement offered, i.e., an AUD50 e-gift card provided as reimbursement upon completion of the interview. Interested participants then provided their consent and completed an online screening survey to confirm their eligibility. Young women aged 16 to 24 years (inclusive) were eligible to participate if they identified as female, lived in Australia, were fluent in English, and endorsed a history of or were currently engaging in self-harm. For participants aged 16–17, the Gillick competency assessment [44] was administered to ensure they had sufficient understanding of what was being proposed. Participants who did not meet the eligibility criteria were redirected to a page that provided information on support helplines. Eligible participants were then required to provide demographic information (such as employment status, education status, and mental health history) and information to facilitate interview scheduling.

### 2.3. Data Collection

Online, one-on-one interviews took place during November to December 2023 via a secure video-conferencing platform (Zoom), using a semi-structured interview guide (see Appendix A) collaboratively developed by researchers, clinical psychologists, and individuals with lived experience of self-harm. The guide included questions about participants’ self-harm history, circumstances surrounding their self-harm behaviour, choice of self-harm methods, participants’ help-seeking behaviour, and their service preferences. While the interview guide did not explicitly ask about peer support, the topic was raised by participants organically, particularly in response to their help-seeking behaviour and service preferences. Five Clinical Psychology Masters students conducted the interviews, each interview lasted approximately 40 to 110 min (average 70 min). Participants were monitored for distress during the interviews and were offered the opportunity to speak with a Clinical Psychologist if they became distressed by the interviews, two participants requested such support at the end of the interviews. Participants were also provided with helpline information at the completion of the interviews. Interviews were audio recorded, de-identified, and then sent to a secure third-party service for verbatim transcription. All audio files sent to the transcription service were permanently deleted after transcription was completed. No identifiable information provided by participants were included in the transcription or reporting of the findings.

### 2.4. Data Analysis

An inductive Reflexive Thematic Analysis [45] was used to develop a broad understanding of peer support received by participants in relation to their self-harm. Analysis was conducted collaboratively by two researchers (AW and DR) using Nvivo Software (version 12). AW was a Clinical Psychology Masters student who was conducting this research as part of her postgraduate studies. She had clinical experience working with individuals who engaged in self-harm. DR was a Ph.D. candidate with extensive experience conducting qualitative research in the self-harm and suicide field. Both researchers’ familiarity with the topic helped to inform their interpretive lens of the data. Both researchers engaged in ongoing reflexive discussions about how their professional backgrounds, assumptions, and proximity to the topic shaped coding and theme development.

After in-depth familiarisation of the transcripts, including repeated reading and note-taking, AW and DR identified all data relating to the participants’ views of peer support for self-harm. This data was then isolated from the remaining transcript data. The isolated peer support data was coded iteratively through line-by-line coding. AW and DR meet regularly during the process to discuss their interpretations of the data, until a final set of codes was developed. Codes were then grouped into themes based on shared meaning. These themes and codes were refined until final themes were created. AW and DR met regularly during the data analysis process to review the coded passages and identified themes with consultation from a third researcher (AH), who provided an additional interpretive lens to facilitate sense making. To support credibility and reflexivity, the research team maintained reflexive notes, files documenting analytic decisions, and continued group discussions throughout the analysis. This process enhanced transparency and ensured that the final themes represented a thoughtful and rigorous engagement with the dataset.

## 3. Results

### 3.1. Participants

The total sample included 27 young Australian women with a mean age of 20.9 years (*SD* = 2.1, range = 18–24). Participants’ demographic and mental health characteristics are displayed in Table 1.

### 3.2. Thematic Analysis

The thematic analysis identified five themes: 1. Peer support offers affirmation; 2. Peer support offers connection to community; 3. Peer support offers empowerment; 4. Capacity of peers impacts helpfulness of peer interaction; and 5. Peer support distorts perception of self-harm. Themes and codes are outlined in Table 2.

#### 3.2.1. Peer Support Offers Affirmation

Participants described peer support playing an important role in affirming young women’s experiences by creating a safe environment where they feel genuinely understood and accepted.

##### Understanding

Participants reported feeling understood in a peer support environment through open and candid discussions about their self-harm experiences. They felt more comfortable sharing their experiences without the fear of being judged negatively. Participants valued having someone with lived experience who truly understood self-harm and mental health struggles, which enhanced the support experience, making it more relatable and effective.

“I think also just it’s a really unique space in that you can talk about it very candidly and people aren’t shocked by it or anything … because there’s a lot of negative self-evaluation about it and just a lot of worry about how they will respond … I can actually talk about this and what I’ve experienced and not have to worry about how people are going to respond because they’ve sort of been there.”—Participant 3

“Having someone to talk to who had been through mental health struggles and actually understood what I was talking about was really helpful.”—Participant 8

##### Validation

Peer bonding through shared experiences provided a space where participants’ personal experiences were validated. Moreover, participants viewed peer support as helpful in reducing their feelings of shame and stigma associated with self-harm. Knowing that others have similar struggles alleviated the burden of feeling different and provided a sense of normalcy, which counteracted feelings of isolation and, in turn, diminished the urge to self-harm.

“It is quite good to talk to someone and share my experiences and have someone in a non-professional sense to validate my experiences.”—Participant 17

“It takes away a bit of the shame around it or feeling like you’re some crazy weirdo.”—Participant 3

#### 3.2.2. Peer Support Offers Connection to Community

Participants discussed peer support as a community, where they experienced connection and acceptance, learnt and grew together, and gained strength from one another’s insights and experiences. In this community environment, collective strength emerged from shared vulnerability and mutual support, turning individual struggles into a source of communal resilience.

##### Connection Through Shared Experience

Connecting with peers was helpful in reducing isolation and promoting positive recovery. Many participants reported they would have welcomed a peer support environment during the time they struggled with self-harm and expressed a need for more accessible and structured opportunities to engage in peer support.

“It would’ve been nice to have a way to connect with peers who were also going through the same thing to take away a bit of that shame.”—Participant 3

“Shared experience is a big thing … a peer support group from similar aged young women who have for one reason or another engaged in self-harm to get together and maybe create a support group for each other … That would’ve been helpful at the time.”—Participant 23

##### Not Feeling Alone

Participants reflected that recognising that they were not alone in their struggles provided a sense of solace and facilitated more effective emotional regulation. The process of sharing their challenges enabled young people to connect and support one another, which enhanced their ability to manage emotions.

“[Working at a Safespace] that’s been really helpful in me realising, okay, I am not the only one who’s been through this self-harm challenges … that’s helped me probably regulate my emotions a little bit better”—Participant 21

“One of my friends knows it’s because she also does self-harm… So it’s something we both struggle with … sometimes being someone, especially at the age that I am, it does feel really isolating. So having someone to run questions past sometimes is nice”—Participant 27

##### Positive Learning from Peers

Participants spoke of the connection with peers allowing them to learn helpful tips and practical strategies in managing self-harm, as well as offering new perspectives and solutions they might not have considered. The shared experience made their advice and suggestions feel more relevant and applicable to participants’ own situations. This peer-driven learning introduced effective coping mechanisms and alternative approaches to dealing with self-harm and related emotions.

“People would talk about what they have found helpful or not helpful and that sort of thing … people would always come up with different things that you haven’t thought of”—Participant 3

“I would love for there to be somewhere where you could really interact with people who are coming out the other end of it … people’s experiences can go a long way in training people and helping people understand things.”—Participant 10

#### 3.2.3. Peer Support Offers Empowerment

Participants discussed that they saw peer support as empowering in several ways. Firstly, by seeing others cope and recover, young women gained hope, confidence, and motivation to pursue their own recovery. Moreover, the autonomy to choose the type of support that resonated with them fostered a sense of agency. Furthermore, taking on supportive roles within peer groups reinforced participants’ own recovery, boosted self-worth, and encouraged accountability and positive change.

##### Hope for Recovery

Seeing others who had developed the skills to reduce and overcome their self-harm behaviours served as a positive indicator that change was possible, which boosted young women’s confidence in their own recovery.

“I think I felt definitely at the times that I felt better, I felt like I was getting better or I was more confident in myself was always when I had seen other adults who had gone through the same thing… You just want to see people on the other side of these things a lot. It’s really hard to see people on the other side of them when you are going through it … I wish I’d had some sort of peer support … I felt like a lot of the supports were difficult to access because I was in a really, really hard time … But eventually I’m seeing other people on the other side”—Participant 10

##### Sense of Control

Participants raised the importance of being able to make choices about their own recovery process. Engaging in a peer support setting, instead of a clinical setting, gave participants agency and control over their recovery journey. They could explore different options, make informed decisions, and take an active role in their healing process.

“I hate sitting in waiting rooms at doctors’ offices and psychologist appointments, and it’s always so formal and so clinical and don’t get me wrong, clinical spaces and clinical things like that have their place. But I also think it’s really great when we can have those non-clinical spaces as well because it gives people more of an option. It lets people choose what they find works best for them … and I guess more control over their own story and over their own life.”—Participant 21

##### Helping Others

Participants described that peer support benefited them by allowing them to take on a supportive role for others, which could be transformative for both the helper and the recipient. This process also offered a unique perspective on participants’ own recovery and seeing the impact of their behaviours on their peers was a powerful motivator to adopt healthier coping mechanisms and fight their urges to self-harm.

“So I work at a Safespace, and I think it’s been really helpful because we encourage visitors to obviously not self-harm and find other ways that are safe to deal with their emotions and regulate their emotions … I can also show that I’ve come out the other side of it too … I can now help somebody else through it. And I think that’s helped me probably regulate my emotions a little bit better and hearing some of the strategies that we teach, try and encourage the visitors to use delaying or distraction … that’s helped a lot”—Participant 21

In addition, one participant mentioned that the mutual recognition of each other’s challenges fosters mutual accountability. Instead of enabling negative behaviours, young people held themselves and each other accountable, enabling a collective commitment to improving their well-being.

“I’ve been in experiences where I’ve had friends who are not doing too good and I was not doing too good. And instead of permitting anything negative that we might do because of that feeling, there is an expectation that we want each other to do better.”—Participant 11

##### Promotes Help-Seeking

Participants discussed that connecting with someone who had personally experienced similar struggles made them more likely to seek help, alleviating the immediate crisis and fostering ongoing engagement with mental health support. The availability of peer support in the community made help accessible and approachable, empowering young people to reach out for assistance without the fear of being judged or misunderstood.

“There’s a really good service that’s quite new … which is just like a walk-in alternative to the ED, for people in mental health crises. And I used that and that was very helpful… You can just walk in and they have peer support workers, which was really good actually because I’m a bit wary of health professionals just because of past experiences. Having someone to talk to who had been through mental health struggles and actually understood what I was talking about was really helpful.”—Participant 8

One participant also reported that she was initially hesitant to disclose her self-harm but confided in a peer who was also self-harming. The friend’s understanding and suggestion to inform a teacher led to the participant seeking further help. This highlighted how informal peer support can encourage young people to seek assistance from trusted adults who were able to direct young people to receive professional mental health care.

“I actually didn’t tell anyone for a really long time. The first person I told, she actually was also self-harming at that point … She responded in quite a gentle way. It wasn’t like, oh, why are you doing that? I think she was understanding because she also was self-harming herself … One of her first responses was, do you want me to tell her [their teacher]?”—Participant 19

#### 3.2.4. Capacity of Peers Impacts Helpfulness of Peer Interactions

Participants raised concerns that young people may not have the knowledge or emotional capacity to provide or receive necessary support. Sharing serious mental health struggles with peers, especially those who were also struggling, could place an undue emotional burden on both parties.

##### Limited Capacity from Young Peers

Participants emphasised that while connection to community and sharing could be beneficial, discussing serious issues like self-harm among young peers without guidance, or relying on peers for emotional support especially when peers were also struggling and lacked the necessary emotional capacity or mental health knowledge, could be harmful and hinder recovery.

“I just had a lot more to learn about relationships and dynamics and how I am and how we all crave community, particularly when we’re 11 to 14. And so when you have a bunch of other 11 to 14 year olds that also are cutting themselves, it’s just such a volatile environment and it’s just not something that I think should be encouraged … it was just not the space to try and rely on people for emotional support when they themselves didn’t have the capacity to provide that and they didn’t have the knowledge that they needed.”—Participant 17

The participant expressed concerns of her own lack of experience or knowledge in providing support.

“When you are talking to other people that are also struggling, you don’t know their triggers, you could trigger someone and you don’t know what.”—Participant 17

One participant voiced her concern that peer interactions may place burden on the helper, negatively impacting their mental health in the process.

“I think even those sorts of friendship groups, which can be in person or online where every person is struggling with mental illness can actually be detrimental. They can be really supportive because you feel understood, but I think ultimately people can end up dragging each other down a bit. And with more online communication… I would stay up late on my phone talking to that partner and trying to convince them not to harm themselves and that affected my sleep, which affected my mental health.”—Participant 1

##### Misinformation and Misuse of Platforms

Participants spoke about the risks of not being able to identify misinformation and the potential misuse of online peer support platforms. Young people may struggle to discern accurate from inaccurate information on social media, which could lead to confusion about what constitutes positive and effective help. Participants felt that more structured and regulated peer support spaces were needed to ensure that the information shared was helpful and that the environment remained supportive and safe for young users.

“If you don’t know the difference between factual and non-factual information, then how would people be able to tell that this is something positive for positive change?”—Participant 7

“(The online app) was more of a community thing … But 12-year-old me didn’t use it in a helpful sense. It was sort of more harmful than it was helpful … other people sharing what they do and then me taking that information on in a way that wasn’t regulated enough.”—Participant 17

#### 3.2.5. Peer Support Distorts Perception of Self-Harm

Participants were also concerned that some peer support engagement may distort perceptions of self-harm by normalising and glamorising the behaviour, or by creating a competitive environment which encouraged taking greater risks.

##### Normalisation and Glamorisation

Participants reported they had seen self-harm being normalised and glamorised within online peer support groups, which reduced the perceived seriousness of the issue, inadvertently motivating and perpetuating more harmful behaviours.

“It’s just such a serious issue that was sort of treated not seriously enough because it was quite normalised.”—Participant 17

“It was something that was very romanticised it … there’s something that kind of puts it on a pedestal and idolises it and to be known as someone that is silently suffering, that feeling of martyrdom was quite powerful”—Participant 20

##### Comparison and Competition

When young women saw that others were self-harming, it could promote a sense of competition to see who could harm themselves more severely, which would exacerbate self-harm behaviours and undermine the recovery process.

“Sometimes I feel like I would’ve spent the whole time comparing myself to other people on how they were hurting themselves and I should be doing it better like them.”—Participant 23

To mitigate these risks, participants outlined that it was important for peer support environments to be regulated with clear guidelines that focus on positive reinforcement and healthy coping strategies.

“it was very much almost like a competition between who was suffering the most. I think that just was not useful. So I think some regulations would probably help quite a lot.”—Participant 22

## 4. Discussion

This study aimed to investigate young Australian women’s views on peer support for self-harm. Five key themes emerged that both align with prior literature and deepen our understanding of how peer support operates in this context. While peer support offers meaningful affirmation, connection to community, and empowerment, it also reveals critical vulnerabilities when peers lack the capacity, training, or support to manage complex dynamics. These findings highlight that lived-experience-driven peer support can carry both promise and risk unless carefully designed and implemented.

A key finding from this study is that peer support offers significant emotional and practical benefits. Consistent with prior research, peer support fosters an environment where young women’s experiences are understood and validated, which helps to alleviate the stigma and shame often associated with self-harm, thereby reducing feelings of isolation [1,24,26]. This aligns with the broader literature suggesting that peer support can mitigate social disconnection, a known factor contributing to self-harming behaviour [1,19].

Another benefit of peer support is the formation of community through shared experiences, helping to restore a sense of connection that often is missing in formal clinical relationships. Supporting prior research, this study found that peer connections promote emotional growth and practical learning, with young women finding strength from common struggles and mutual understanding [23,27,31]. The sense of acceptance and belonging cultivated through peer interactions plays a critical role in diminishing young women’s urges to self-harm by alleviating isolation and promoting self-acceptance, which, in turn, enhances their self-efficacy in dealing with personal challenges [46].

Moreover, peer support empowers young women by promoting hope for recovery, autonomy, and help-seeking behaviour, and providing opportunities to help others. Firstly, consistent with the literature, observing the recovery of peers was found to instil hope and reinforce a sense of control over young women’s own recovery, and thereby increasing their confidence in overcoming self-harm urges [25,32]. More importantly, this study found that peer support enabled young women to explore various approaches in managing self-harm, balancing peer-led versus clinical interventions based on their personal goals, and make informed decisions suited to their unique circumstances. In this way, peer support could act as a bridge, helping translate clinical strategies (e.g., DBT, CBT skills) into everyday coping, reinforcing them through lived-experience credibility and ongoing relational support. This complementary role is consistent with integrated peer-clinical models in broader mental health services, i.e., with peer support workers embedded alongside clinicians and social workers [47]. Emerging evidence of optimal integration of such models has been reported in recent studies [48,49]. Our findings suggest that support initiatives for self-harm will be more effective when designed with flexibility, allowing users to engage in ways that best suit their emotional needs, personal goals, and capacities.

Furthermore, this study found that by supporting others, young women remind themselves of the importance of seeking healthy alternatives to self-harm. Knowing that they can make a positive difference in someone else’s life also reaffirms their values beyond their struggles. These findings add support to the wider peer support literature, which suggests that helping others within the peer support environment is associated with improvements in the help providers’ emotional regulation, confidence, and self-worth [21]. Overall, the opportunities to share their experiences, learn from others, and contribute to a supportive community provide young women with a renewed sense of purpose, which is crucial in their personal recovery journey [50].

While the benefits of peer support are significant, our findings also highlight substantial limitations tied to peers’ capacity. A key concern that emerged from this study is that young peers often lack the skills or experiences needed to provide or receive meaningful support. This finding aligns with a recent qualitative study in which young people reported feeling unprepared and lacking sufficient skills when supporting friends who also engage in self-harm [28]. Consequently, although validation, connection, empowerment are notable strengths of peer support, the impact can vary—peer support may foster positive changes or inadvertently reinforce maladaptive behaviours, depending on the knowledge, skills, and emotional capacity of the individuals involved. For example, young peers may unintentionally reinforce harmful behaviours by validating maladaptive coping mechanisms; or if their empowerment stems from unhealthy practices, such as viewing self-harm as a form of personal control, the messages they share through peer interactions would be counterproductive; moreover, young peers may become emotionally overwhelmed by the burden of supporting someone else while still grappling with their own mental health challenges.

The current findings offer a possible explanation for the mixed outcomes reported in previous research that while exposure to triggering self-harm content was found to be associated with increased self-harm behaviour for some individuals, it has also been shown to lead to resistance of self-harm urges by others [37,38,39]. It is likely that individuals with greater mental health knowledge, stronger emotional regulation, and higher motivation for recovery interpret such exposure as a deterrent. Confronting others’ self-harm behaviours increases their awareness of the associated negative consequences and reinforces their commitment to avoid further self-harm. Recognising that the effectiveness of peer support depends not only on the shared content or peer dynamics, but also on the capacity of those involved, again highlights the need for knowledge, skills and emotional resilience building for young people engaged in peer support settings.

The other concerns reported in this study also align with previous research, highlighting the risks of poorly moderated peer groups, especially in online settings. In un-regulated spaces, young users may encounter inaccurate or harmful advice, and self-harm behaviour risk being normalised, glamorised, or even promoted through competitive dynamics [31,33,36]. While these groups aim to provide safe spaces for peer-to-peer interactions, they may inadvertently shift focus away from recovery by reinforcing unhealthy comparisons and beliefs.

Compared to the broader mental health literature, for example, studies on substance use disorder, where peer support interventions have proven to be effective in promoting recovery outcomes [51,52], it is important to note that successful peer support groups are typically structured and facilitated by trained peer moderators [53]. These peer moderators receive ongoing guidance from professional mental health clinicians and are trained to de-escalate crisis situations and redirect users to professional help if appropriate [53]. The current findings support this approach, as young women expressed a desire for moderated peer support environments to ensure peer-led interactions remain focused on promoting healthy, evidence-based recovery strategies.

Findings from this study suggest that psychoeducation around self-harm, trainings in communication skills, effective and safe sharing of lived experiences, and boundary management would help young women engage more safely and effectively in peer support interactions. The study results also highlight the need for more structured and regulated peer support programs or online space, for example, including peer moderators, to ensure user safety and mitigate unintended harm. Moderator training could include core components such as psychoeducation about self-harm, recognising early warning signs of distress, boundary-setting practice, and escalation procedures with increased risk. In the online space, clear content guidelines, including appropriate language use, proactive trigger warning, and restrictions on graphic or comparative self-harm disclosures are essential. Content moderation is critical to prevent distorted narratives of self-harm and the spread of inaccurate and harmful information. Trained peer moderators could play a vital role in facilitating this process, and they should also be supported by clear escalation and referral protocols, outlining when and how to involve clinical services if safety concerns arise. These insights warrant the design of peer-led programs and spaces with structured training, moderation, and clear pathways to clinical involvement when needed. Future studies could evaluate such models using outcome measures such as user acceptability and level of user engagement.

Findings of the current study are subject to several limitations. First, the interview process was developed for a broader research initiative rather than with a specific focus on peer support. As a result, peer support for self-harm was explored primarily within the context of help-seeking behaviours and service preferences, limiting the depth of exploration with participants. This may have reduced the transferability of the findings, and as such, future research should consider examining this phenomenon in more detail. Secondly, it is important to acknowledge that only a subset of participants (nine out of 27) had direct experience engaging in peer support at the time of the interviews. The remaining participants discussed peer support opportunities they wished had been available to them or their perceptions of what might be beneficial to them. This may reflect the scarcity of peer support opportunities available or participants’ lack of awareness of such programs, neither of which were explicitly explored in this study. Nonetheless, this variation in experience may influence the depth and specificity of insights provided. It is possible that participants’ views may change if they had been personally involved in peer support interactions. Consequently, the findings may not fully capture the perspectives of young women actively engaged in peer support for self-harm. Finally, this study adopted a broad definition of peer support, encompassing diverse formats, including informal peer interactions, structured peer work programs, as well as online and offline communities. While this inclusivity provides a comprehensive overview, treating peer support as a broad contrast may limit comparability across settings and constrain the extent to which findings can be generalized to specific peer support models. Future studies would benefit from exploring peer support subtypes more systematically, including how the structural, relational, or technological features of peer support influence perceived benefits and risks, to deepen our understanding of their unique contributions to young people’s recovery in self-harm.

## 5. Conclusions

This study adds to the limited body of qualitative research exploring young women’s views on peer support for self-harm. Our findings suggest peer support for self-harm has the potential to fill critical gaps, especially for young people not connected with formal services, by providing relational and experiential support. With appropriate safeguards, such as incorporating capacity training, and including trained peer moderators, peer support can be a powerful complement to existing mental health services. For clinical practice, these findings suggest that clinicians working with young women who self-harm may benefit from sensitively exploring whether and how peer support features in a client’s coping repertoire. Clinicians may also consider integrating guided discussions about safe peer-support engagement into follow-up care.

## Figures and Tables

**Table 1 ijerph-22-01874-t001:** Demographic and mental health characteristics of participants.

Demographic		*N* (%)
Employment status	Part-time	21 (78%)
	Unemployed	6 (22%)
Education status	Full-time university	10 (37%)
	Part-time university	6 (22%)
	Part-time vocational training	2 (7%)
	Not currently studying	9 (33%)
Mental health diagnosis (not exclusive)	Mood disorders	21 (78%)
	Anxiety disorders	17 (63%)
	Eating disorders	7 (26%)
	Neurodevelopmental disorders	7 (26%)
Self-harm behaviours	Currently engaged	15 (56%)
	No longer engaged	12 (45%)

Note: *N* = 27.

**Table 2 ijerph-22-01874-t002:** Theme structure.

Themes	Codes
3.2.1. Peer support offers affirmation	Understanding
	Validation
3.2.2. Peer support offers connection to community	Connection through shared experiences
	Not feeling alone
	Positive learning from peers
3.2.3. Peer support offers empowerment	Hope for recovery
	Sense of control
	Helping others
	Promote help-seeking
3.2.4. Capacity of peers impacts helpfulness of peer interaction	Limited capacity from young peers
	Misinformation and misuse of online platforms
3.2.5. Peer support distorts perception of self-harm	Normalisation and glamorisation
	Comparison and competition

## Data Availability

The data presented in this study are available on request from the corresponding author due to ethical considerations.

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
