# Peer review of "Young Australian Women’s Views on Peer Support for Self-Harm: A Qualitative Study"

_ijerph, 2025, doi:10.3390/ijerph22121874_

Round 1

Reviewer 1 Report

Comments and Suggestions for Authors

The authors present very powerful work based on interviews with young women in Australia who have a history of self-harm. Please see few suggestions below that may help improve this work/manuscript.

I am aware that deliberate self-harm is a term and keyword used in this area of work for many years now and often journals ask authors to add keywords in their submission to make it more easily discoverable on web sources. I would suggest using a wider keyword term such as self-injurious behaviour or intentional self-harm and omitting deliberate self-harm as some lived experience people do not approve this term anymore. Self-injurious behaviour or intentional self-harm are part of the Mesh Term so your study will also be discoverable.

Introduction

The introduction is well written and informative. The authors are focusing on women given the increasing self-harm rates, but it will be also important to cite some studies, qualitative, that have focused on men/male self-harm experiences. This will be helpful for the readers. My suggestion is not about comparing females to males but to give a wider picture of both females/males and may help future studies.

Recruitment

The authors mention on the data collection the e-gifts given to participants, but it is better to use this in the recruitment section. Was the e-gift card mentioned on the survey/website, consent forms? Was the price mentioned? Good to mention these details in this section.

Furthermore, for this section, was it lived experience or living experience of self-harm you were focusing on? Is there a difference? Good to clarify it in this section as  I see you use both terms in general.

Data collection

Good to mention that the audio records sent to the third-party service were deleted after being transcribed, i.e. how long after the transcription, 1 week for example?

Within the process, recruitment and data collection, can the authors add a statement regarding reflexivity that is usually included in qualitative studies based on interviews? In addition, was there a PPI person employed for this work and did they provide their input for recruitment or manuscript written? If no PPI person was involved, better to state that in the methodology.

Conclusions

Can the authors add something about further studies? What does your current findings highlight  for next steps? What can a clinician keep in mind for their practice and next care follow up for women presenting with self-harm to mental health services. Has peer support been used for other populations? For example, First Nation people often have peer-support groups, has this been helpful for suicidal behaviours?

Reviewer 2 Report

Comments and Suggestions for Authors

Thank you very much for this important and interesting study. It is performed and written clear and sound. My only comment is that it is exclusively about women. I understand why, the increasing self-harm among women, but what about men? It could be interesting, for instance in the discussion, to elaborate a bit about men and self-harm, would it be very different from women, especially in peer support? 

Reviewer 3 Report

Comments and Suggestions for Authors

Dear authors

After reviewing your manuscript,  I suggest implementing the following changes:

1.- Please, ensure the abstract explicitly states the qualitative design (reflexive thematic analysis), the sampling approach, and the primary thematic domains (benefits/risks), and avoids implying effectiveness or causal claims.

2.- Clarify alignment between aims, interview guide, and analysis. State explicitly that the focus on “peer support” emerged within a broader study, describe how this analytic shift was tracked (audit trail), and explain implications for scope and transferability in Methods and Limitations.

3.- Strengthen qualitative rigour and reflexivity. Expand on coder positionality (training, professional background, relationship to the topic), describe credibility procedures (member checking, peer debriefing, triangulation, reflexive journaling as applicable), and justify sample adequacy using an information power or saturation argument with citation.

4.- Enhance sample description and transferability. Report key contextual characteristics (e.g., geographic distribution, socioeconomic background, cultural identity), and state clearly how many participants had direct experience with peer support. Add a supplemental table with participant characteristics.

5.- Deepen the analytic procedure. Describe in detail how codes were generated and aggregated into themes, how disagreements were resolved, whether NVivo (or similar software) supported coding, and include negative/discordant cases. Consider adding a quotations column to an existing table or a separate quotes table by theme.

6.- Define “peer support” more precisely.

Distinguish informal vs. formal peer support and moderated online vs. unmoderated spaces—anchor recommendations to each subtype across the Introduction and Discussion/Implications.

7.- Make implications operational. Specify minimum moderator training components (psychoeducation, boundary-setting, risk escalation), content guidelines (language, trigger warnings), escalation/referral protocols, and propose outcome measures for future feasibility pilots (acceptability, fidelity, adverse events, engagement).

8.- Expand Limitations. Note that only a subset of participants reported direct peer-support experience, and the breadth of the “peer support” construct limits comparability across settings.

Recommend future work focusing on specific peer-support models (e.g., moderated in-person groups vs. online communities).

Reviewer 4 Report

Comments and Suggestions for Authors

The manuscript under review is devoted to the important and practically significant problem on self-harm among young people among young people (on example of young Australian women). The purpose and content of the article correspond to scope of the journal IJERPH.

The research presented in the article makes a good impression in its design and implementation.

The authors clearly stated the study's purpose and described its design and qualitative methods in detail. The study's results are presented fully and discussed thoroughly, and the limitations of the study are outlined.

I have only a few recommendations for improving the article:

  • The authors need to provide a more detailed justification for why the study only examines young women. The authors note in the text that "The rising rates of self-harm, particularly among young women, are alarming" (lines 39-40). However, this is insufficient to explain why it is important to study this issue among young women rather than young men. This aspect should be further justified and/or noted in the study's limitations
  • The authors indicate that a semi-structured interview guide is presented in Appendix A, but the article text does not contain any appendices. An appendix should be added to the text, or the guide should be indicated in the supplementary materials. The first option is preferred (for the convenience of future readers), but this is at the discretion of the editorial board
  • The authors note that "the risks of poorly moderated peer groups, especially in online settings. In unregulated spaces, young users may encounter inaccurate or harmful advice, and self-harm behavior is at risk of being normalized, glamorized, or even promoted through competitive dynamics" (lines 536-538). However, it can be assumed that this risk exists not only due to poor group moderation; most likely, it is a manifestation of protective mechanisms. In this regard, in my opinion, it would be advisable for the authors to more fully describe the possibilities of peer support for the prevention of self-harm.
